# PREFERENCE DATA ANNOTATION WITH GUIDED DENSITY RATIOS

## ABSTRACT

Preference tuning of large language models (LLMs) relies on high-quality human preference data, which is often expensive and time-consuming to gather. While existing methods can use trained reward models or proprietary model as judges for preference annotation, they have notable drawbacks: training reward models remain dependent on initial human data, and using proprietary model imposes license restrictions that inhibits commercial usage. In this paper, we introduce Guided Density Ratio, a *training-free and highly effective* method that leverages off-the-shelf LLMs for preference data annotation. Our approach uses the log-density ratio between a better-aligned LLM and a less aligned LLM as a reward signal. We explores 221 different LLMs pairs and empirically demonstrate that increasing the performance gap between paired LLMs correlates with better reward generalization. Furthermore, we show that tailoring the density ratio reward function with specific criteria and preference exemplars enhances performance across domains and within target areas.

In our experiment using density ratio from a pair of Mistral-7B models, Guided Density Ratio achieves a RewardBench score of 82.6, outperforming the best trained reward functions from same model class and demonstrating competitive performance against SoTA models in Safety (91.0) and Reasoning (88.0) domains. We use Guided Density Ratio to annotate an on-policy preference dataset with which we preference tune *Llama-3-8B-Instruct* with SimPO. Using reward signals from two relatively weak models, our approach pushes Llama-3-8B to achieve a 37.4% (+15.1%) win rate on ArenaHard and a 40.7% (+17.8%) win rate on Length-Controlled AlpacaEval 2.0, along with a score of 8.0 on MT-Bench.

## 1 INTRODUCTION

Preference tuning has advanced the capabilities of large language models (LLMs), but this progress relies on high-quality human preference data which is both costly and time-consuming to gather. Cutting-edge models (e.g., ChatGPT, GPT-4, Claude-3) are aligned with curated, quality-controlled human preference data, typically provided by specialized companies. While effective, this approach limits broader adoption due to prohibitive costs and limited transparency in data collection (Wang et al., 2024c). AI-feedback solutions are emerging as an alternative—either through a trained reward model (Dong et al., 2024) or proprietary LLM-as-a-judge (Cui et al., 2023). However, training reward models still rely on costly initial human preference data, and proprietary LLM-as-a-judge approaches introduce licensing restrictions that generally prevent commercial use.

This paper introduces Guided Density Ratio that leverages the density ratio between off-the-shelf LLMs to efficiently facilitate preference data annotation. Our method uses the log-density ratio between a better-aligned model and less-aligned model to annotate preference data. We show that a higher alignment gap between model pairs yields improved preference signal. This observation, referred to as the "Strong-over-Weak Hypothesis", is supported by our experiments across 221 model combinations (Figure 1). Notably, log-density ratios between a post-DPO model and a pre-DPO model, known as the DPO implicit reward (Rafailov et al., 2023), have not gained widespread adoption due to limitations in generalizability and high reward variance in reward performance across different model choices (Lambert et al., 2024; Lin et al., 2024). We show that the *implicit DPO reward is an empirically suboptimal special case of the strong-over-weak density ratio reward,*

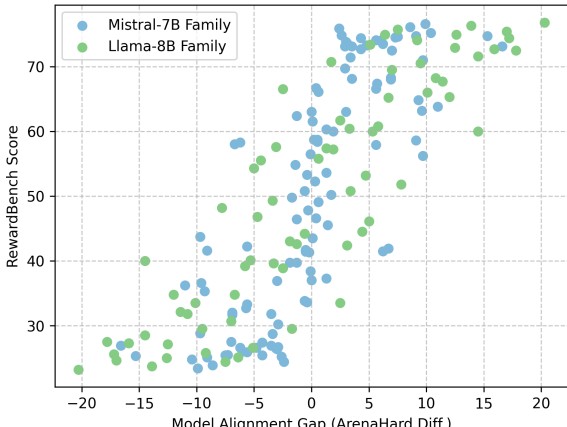

Figure 1: Scatter plot of model alignment gap (x-axis) versus RewardBench score (y-axis) for models in the Llama-8B (orange) and Mistral-7B (blue) families. Models' human-alignment gap is measured using difference in ArenaHard scores between $\pi_{\text{strong}}$ and $\pi_{\text{weak}}$ policies belonging to the same model family. Each point represents one of 221 unique model pairings from these families (10 models from Llama-8B, 11 from Mistral-7B). Models include Base, SFT, and preference fine-tuned with algorithms such as DPO, PPO, KTO, RRHF, ORPO, SimPO, IPO, and SLiC-HF. The plot demonstrates a positive correlation between model alignment gap and RewardBench score, indicating that higher alignment gaps correspond with more effective reward model performance.

with its performance variance reducible by consistently choosing a weaker model as the reference (Figure 2). Our findings highlight the importance of selecting model pairs with a sufficient alignment gap and demonstrate the flexibility to construct the density ratio reward using models trained with various objectives (such as SFT, RRHF, SLiC-HF, ORPO, SimPO, KTO, IPO, etc.)

We guide our density-ratio reward function to align with the domain of each sample in the annotation set. Given that human preferences span multiple dimensions (e.g., trustworthiness, reliability), an effective reward function should adapt to requirements specific to each domain. Guided Density Ratio introduces an end-to-end process, from identifying the domain of each user query to customizing the reward function to prioritize the relevant preference criteria. Specifically, Guided Density Ratio first uses an adaptive router to identify the domain of each user query (e.g., chat, reasoning, safety). It then applies domain-specific instructions and in-context learning (ICL) examples to clarify preference criteria. In this way, we customize a density-ratio reward function from a general preference signal to domain-specific annotators. Experimental results demonstrate that adaptively guided density ratio significantly improve in both overall and target domain reward generalization.

The main contributions of this paper are as follows.

- **Choosing Models via Strong-over-Weak Hypothesis**. We propose a *training-free* framework to leverage the density ratio between a better-aligned LLM and a less-aligned LLM as a reward signal for annotating preference data. We introduce the "Strong-over-Weak Hypothesis", which suggests increasing the preference gap between the two LLMs to improve the accuracy of the density ratio reward function. Through extensive experimentation on 221 model pairs, we empirically validate this hypothesis. Our findings can reduce the reward variance seen in existing density ratio methods (e.g., DPO Implicit reward) and offer guidance on selecting effective model pairs for density ratio-based reward functions.

- **Customizing Reward Function via Prompting**. We customize the density ratio reward function for target domains by incorporating domain-specific instructions and ICL examples. Our experiment on RewardBench shows significant domain-wise improvement after applying prompt guidance: Safety domain improved from 82.4 to 91.0, the Reasoning domain performance from 73.8 to 88.0, and the ChatHard domain from 60.4 to 69.7. Guided Density Ratio uses a LLM-based router to assign customized instructions for different examples in the annotation set.

The adaptive guided reward function improves over density ratio reward w/o prompting by 5.3 points overall on RewardBench, and exceeds best trained classifier of same model class.

- **Alignment Improvement**. We use Guided Density Ratio of a pair of Mistral-7B models to annotate on-policy preference data collected through Best-of-N sampling. Training on this data, the *Llama-3-8B-Instruct* model is aligned to achieve a 37.4% (+15.1%) win rate on ArenaHard, a 40.7% (+17.8%) length-controlled win rate on AlpacaEval 2.0, and a score of 8.0 on MT-Bench, reaching performance comparable to SoTA trained reward functions.

## 2 RELATED WORKS

**Density ratios for alignment**    Density ratio as rewards is popularized as implict DPO reward (Rafailov et al., 2023). Chen et al. (2024) uses implicit DPO reward to bootstrap an LLM through iterative DPO training. Zhong et al. (2024) trains a DPO model and uses the density ratio to derive a token-level characterization for response quality, and uses it as a reward signal in PPO training. Yang et al. (2024b) uses the density ratio between DPO vs SFT model as quality filter. Though one study Lin et al. (2024) finds that implicit DPO reward struggles to generalize on OOD examples compared with just training a classifier using (BradleyTerry; Bradley & Terry, 1952) objective. This paper shows that implicit DPO reward is only a special case in the class of strong-over-weak density ratio reward. Our findings provide guidelines to identify suitable pairs to construct DR. In particular, the denominator model must be sufficiently weak and unaligned to contrast with numerator model to provide generalizable preference signal.

**Discriminative & generative preference**    Trained classifiers and generative rewards are the existing approaches for preference data annotation. They top leaderboards such as RewardBench  (Lambert et al., 2024) and are widely adopted by both industry and the community to preference align well-known models (Ouyang et al., 2022; Touvron et al., 2023; Adler et al., 2024; Yang et al., 2024a; Cui et al., 2023).  In fact, due to the scarcity and noise of human preference data, GPT-4 based generative rewards have long been harvested to align models, such as reinforcement learning from AI feedback (RLAIF; Bai et al., 2022). High quality and popular preference datasets are actually often times annotated by LLM-as-a-judge, both in the forms of scalar score and textual assessment and critiques (Cui et al., 2023). With high quality human data or LLM generated data available, one can fine-tune a LLM to be a better generative judge (Wang et al., 2024b; Zhang et al., 2024; Wang et al., 2024a; Kim et al., 2024), or to tune a linear layer on top of the LLM to be a sequence classifier (Adler et al., 2024; Dong et al., 2024; Liu & Zeng, 2024). However, such approaches either require quality data for training or a powerful closed-source LLM which may be prohibitive for license concerns.

**Weak-to-strong generalization**    Many works have explored the idea of using a weak and a strong model to obtain better performance than the strong model.  Contrastive decoding, for instance, enhances LLM generation quality by searching for sequences that maximizes the likelihood different between an expert model and an amateur model. O'Brien & Lewis (2023) shows CD consistently improves reasoning tasks.  Li et al. (2022) shows improved generation quality in wikipedia, news and story domains. Chuang et al. (2023) shows improvement in LLM facutuality by contrasting the differences between logits in later layers and earlier layers. In addition to contrastive decoding, EXPO (Zheng et al., 2024) is a model extrapolation method that leverages the delta between an aligned model and pre-aligned model as a global gradient update to the aligned model, yielding surprising improvement over various evaluation benchmarks. Burns et al. (2023) found that using supervision from a weak model to train a strong model can yield better-than-supervisor performance from the stronger model.

## 3 METHOD

We study two research questions critical to reward function design based on density ratio. First, we discuss how to find successful pairs of LLMs to construct effective density ratio reward (section 3.1). Our results reveal a strong correlation between the alignment gap of model pairs (measured by the ArenaHard score) and the effectiveness of the reward function (evaluated through the RewardBench score). Second, we study how to use prompt guidance to make density ratio reward customizable to human defined criterion (section 3.2).

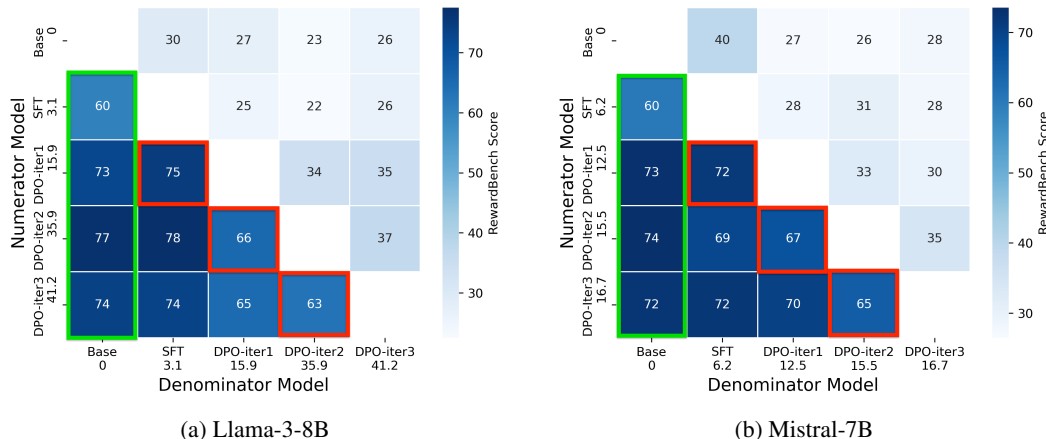

(a) Llama-3-8B  (b) Mistral-7B

Figure 2: Density ratio reward from different pairing combinations, with y-axis the numerator model, and x-axis denominator model. The five models chosen in each model family are sorted by their human-aligned level measured by ArenaHard. According to DPO implicit reward theory, models along the diagonal (red-outlined cells) theoretically yield optimal rewards, pairing models before and after DPO training. However, empirical results indicate that using the Base model as the denominator consistently yields higher scores (green-outlined cells), motivating our strong-over-weak density ratio reward function.

## 3.1 REWARD FUNCTION DESIGN USING LOG-DENSITY RATIO

**Motivation**    We explore constructing density ratio reward function with various pairings of LLMs. According to DPO implicit reward theory (see Section 5.1 in Rafailov et al. (2023)), optimizing DPO objective also learns an implicit reward function $r(x,y) = \beta \log \frac{\pi_\theta(y|x)}{\pi_{\text{ref}}(y|x)}$ that approximates the ground-truth reward. This function is named DPO implicit reward, and follow-up works (Lambert et al., 2024; Lin et al., 2024; Chen et al., 2024) have directly leveraged it to annotate preference data.

To assess this theory, we conduct an experiment applying online iterative DPO (Xiong et al., 2023; Xu et al., 2023; Swamy et al., 2024) to both the Mistral and Llama-3 model families. The key ideas of online iterative DPO training are: (1) the reference model is updated at each iteration (i.e., $\pi_{\text{ref}} = \pi_{\theta_{t-1}}$), and (2) the training data is also updated iteratively, with $y_w$ and $y_l$ generated by sampling from $\pi_{\theta_{t-1}}(\cdot \mid x)$ and annotated with preferences using an external reward function. This process starts with an SFT model trained from Base, and iteratively trains three DPO checkpoints from 3 DPO optimization iterations (checkpoints from Chen et al. (2024)). The DPO model ($\pi_{\theta_t}$) at iteration $t$ uses the policy model at the previous iteration ($\pi_{\theta_{t-1}}$) as its reference model. We write the loss function of iterative DPO loss function in (equation 1):

$$\mathcal{L}_{\text{iter\_DPO}}(\pi_{\theta_t}; \pi_{\theta_{t-1}}) = -\mathbb{E}_{(x,y_w,y_l)\sim\mathcal{D}_t}\left[\log\sigma\left(\beta\left(\log\frac{\pi_{\theta_t}(y_w \mid x)}{\pi_{\theta_{t-1}}(y_w \mid x)} - \log\frac{\pi_{\theta_t}(y_l \mid x)}{\pi_{\theta_{t-1}}(y_l \mid x)}\right)\right)\right].$$
(1)

We ensure that the DPO objective is effectively optimized at each iteration, as we see consistent improvement in updated model's ArenaHard score (see Figure 2). Then, according to the implicit DPO reward theory, one would expect the density ratio between $\pi_{\theta_t}$ and $\pi_{\theta_{t-1}}$ to provide an optimal reward function. However, Figure 2 shows that using weaker models–such as the base or SFT models—as the denominator in the log-density ratio, rather than $\pi_{\theta_{t-1}}$, leads to significantly better reward functions evaluated by RewardBench. This result shows that empirically, DPO implicit reward is suboptimal compared with simply choosing weaker reference policies, implication of which motivates us to propose "Strong-over-Weak Hypothesis" to guide model pairing.

**Reward Function Design** We use the following reward function to annotate preference data.

$$r(x, y) = \log \frac{\pi_{\text{strong}}(y \mid x)}{\pi_{\text{weak}}(y \mid x)} = \log \pi_{\text{strong}}(y \mid x) - \log \pi_{\text{weak}}(y \mid x). \tag{2}$$

Here $\pi_{\text{strong}}$ and $\pi_{\text{weak}}$ are two off-the-shelf LLMs from the same model family with $\pi_{\text{strong}}$ outperforming $\pi_{\text{weak}}$ across all dimensions of human preference, such as safety, correctness, and relevance.

In Section 4.1, we conduct extensive experiments on density ratio reward involving 221 distinct model pairs. Our findings reveal a strong correlation between the alignment gap of $\pi_{\text{strong}}$ and $\pi_{\text{weak}}$ and the effectiveness of the reward function, as quantified by the RewardBench score. As shown in Figure 1, *achieving an effective reward function in Eq. (2) necessitates a substantial human-alignment level difference between $\pi_{strong}$ and $\pi_{weak}$*. We term this insight the "Strong-over-Weak Hypothesis" , which can serve as a guiding principle for selecting models in density ratio reward computation. Furthermore, our experiments span a range of models fine-tuned using different preference tuning strategies (e.g., DPO, SimPO, KTO, ORPO) to show that density ratio reward is not exclusive for DPO models, with further details provided in Figure 5. We summarize our key observations below.

- We recommend using a weak model for the denominator in (2) that has not been fine-tuned on human preference data, such as an *SFT* or *base* model. For the numerator, a stronger model that aligns more closely with human preferences (e.g., AlpacaEval2.0 or ArenaHard benchmarks) should be used. This approach maximizes the performance gap, often leading to better performance of the reward function.

- We recommend using both strong and weak models from the same model family. If the weak model is an SFT model, we suggest using a strong model that has been preference-tuned from this SFT model. This approach ensures that when leveraging existing benchmarks (e.g., AlpacaEval 2.0 or ArenaHard) to evaluate the performance gap in human preference alignment, potential confounding factors, such as differing inductive biases between unrelated models, are minimized.

---

You are a helpful AI assistant. You follow the following guidelines when answering user questions.

**1. Answer Constructive, Clear Questions**
- Provide an answer when the user asks for factual information, constructive advice, or help with personal growth. Focus on offering practical, positive guidance.

**2. Recognize Jokes, Puns, and Fictional Contexts**
- Respond playfully when the question references humor, games, movies, or fictional scenarios. Acknowledge the fictional nature while keeping the tone light.

**3. Avoid Answering Harmful, Illegal, or Malicious Questions**
- Do not engage if the question promotes harm, illegal activities, or unethical behavior. Politely but firmly refuse to provide an answer, while keeping the response respectful.

**4. Handle Sensitive Topics with Empathy**
- Respond with care to questions about mental health, personal relationships, or emotionally charged situations. Acknowledge the user's feelings, and offer general advice or suggest professional resources.

---

Figure 3: Instruction with detailed criterion to define preference in Safety domain. This prompt outlines key principles to ensure constructive, empathetic, and safe responses.

## 3.2 GUIDED DENSITY RATIO

Human preferences are multi-dimensional (e.g., safety, trustworthiness, reliability, faithfulness) (Bai et al., 2022; Wang et al., 2024c; Naseem et al., 2024), and an effective reward function should adapt its criteria according to the specific domain requirements. For example, a chatbot explaining corporate vacation policies should emphasize faithfulness to company policy and the accuracy of its responses, rather than focusing on aspects like conversational style or user engagement. However, vanilla log-density ratio reward function provides a single, aggregated reward signal, merging various, potentially conflicting preference aspects. Therefore, it is crucial to define preferences clearly and

provide concrete examples to tailor the contrastive reward signal to the specific domain or aspect of human preference.

We introduce Guided Density Ratio, which specifies the domains for each prompt and incorporates instructions and in-context-learning (ICL) examples to define criteria for positive and negative preferences (see figure 4). Each domain has customized instructions and ICL examples, and we ensure diversity by preparing multiple ICL demonstrations, sampling one randomly for each instruction. Formally, for each original user prompt $x$, we inject ICL examples and domain-specific instructions $\mathrm{T}(x)$ to guide the annotation toward relevant preference dimensions. This is equivalent to adapting the reward function into the following form, incorporating $\mathrm{T}(x)$ before applying the log-density ratio for annotation.

$$r_{\text{Guided Density Ratio}}(x, y) = \log \pi_{\text{strong}}(y \mid \mathrm{T}(x), x) - \log \pi_{\text{weak}}(y \mid \mathrm{T}(x), x). \tag{3}$$

To automate annotation, we introduce a domain router that identifies the most relevant domain for each user query, allowing us to apply the appropriate preference criteria for each example in annotation set. For instance, a sensitive query is routed to a Safety expert, while a math or coding query goes to a Math/Code expert. In this paper, we employ the Mixtral 8x7B Instruct v0.1 model (Jiang et al., 2024) with zero-shot prompting to classify prompts into predefined categories (e.g., safety, reasoning, chat) based on a system prompt and task description.

These in-context examples and instructions serve as both demonstrative and descriptive tools to help define and refine the model's preference criterion. Example templates we used can be found at figure 3 and figure 4. For domains like safety, instructions should include guidelines on how to avoid risky outcomes, while in domains like math, demonstrating the preference criterion through examples may be more effective. These instructions provide high-level guidance by defining overarching principles or definitions that shape the reward function's preferences during data annotation.

```
<|im_start|>system
{system_prompt}
<|im_end|>

<|im_start|>user
You must carefully understand my question and give a relevant, cor-
rect, and logical answer.

For example:
User:  {ICL_query_i}
Good Assistant:  {chosen_response_i}
Bad Assistant:  {rejected_response_i}
Explanation:  {explanation_i}

User:  {user_query}
Good Assistant:
<|im_end|>
```

Figure 4: Few-shot Instruction template to guide rewards.

## 4 EXPERIMENTS

### 4.1 STRONG-OVER-WEAK DENSITY RATIO REWARD

**Setup** We collect off-the-shelf LLMs from Mistral and Llama model family with varying levels of human alignment. We use ArenaHard (Li et al., 2024) benchmark (see A.3) to approximate each model's degree of alignment to human preference. We then test distinct pairs of models' density ratio reward by evaluating on RewardBench (Lambert et al., 2024). Each test sample in RewardBench consists of human verified pairwise responses, one chosen and one rejected. The reward function then assigns annotations by comparing the density ratio scores of these two responses. The final RewardBench score reflects the accuracy of a reward function's predictions against human-annotated ground truth.

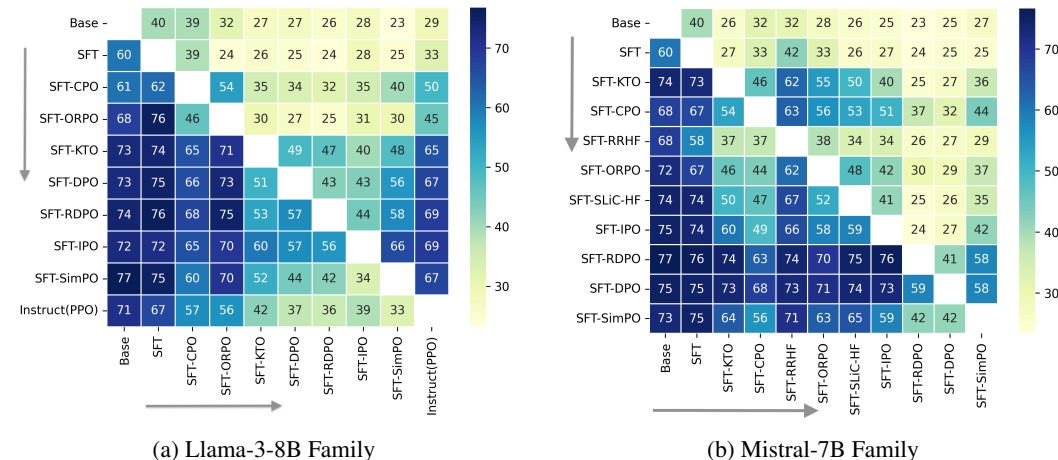

(a) Llama-3-8B Family        (b) Mistral-7B Family

Figure 5: Density ratio rewards from various numerator and denominator model pairings, following Equation (2). Models, fine-tuned with different objectives, are ordered by their human-aligned levels measured by ArenaHard. Generally, larger alignment gaps between numerator and denominator models yield stronger reward functions, supporting the "Strong-over-Weak Hypothesis" in our reward design. This trend holds across models fine-tuned with distinct objectives. An exception, Instruct(PPO)—an official Meta instruct model—achieves a strong ArenaHard score likely due to more intensive SFT training rather than improved human alignment.

Our experiment includes base models, supervised fine-tuned (SFT) models, as well as models optimized through various preference-tuning algorithms (mostly obtained from Meng et al. (2024)). These models are respectively labeled as SFT-objective-name in Figure 5 and span techniques such as KTO (Ethayarajh et al., 2024), CPO (Xu et al., 2024), RRHF (Yuan et al., 2023), ORPO (Hong et al., 2024), SLiC-HF (Zhao et al., 2023), IPO (Azar et al., 2023), RDPO (Park et al., 2024), DPO (Rafailov et al., 2023), SimPO (Meng et al., 2024), and PPO (Ouyang et al., 2022).

**Results** Our findings, visualized in Figure 1, reveal a strong correlation between the accuracy of the reward function in Equation (2) and the strong-over-weak alignment gap. As the alignment gap widens, the reward function achieves stronger results. When the alignment gap is near zero, the signal becomes noisy, with the RewardBench accuracy approximating 50%, indicative of a random guess. Further details are presented in Figure 5, where each row represents a numerator model and each column a denominator model. Each cell displays the paired density ratio reward function's RewardBench score. The heatmap illustrates that the choice of denominator model substantially influences reward generalization; more effective and stable reward functions can be obtained by selecting weaker denominator models (e.g., Base or SFT) to ensure a sufficient alignment gap.

The finding also shows considerable flexibility in constructing density ratio reward. For instance, as shown in Figure 1 (left), SFT-RDPO as the numerator performs well with various checkpoints—such as Base, SFT, KTO, RRHF, SLiC-HF, and IPO—as denominators, producing high reward accuracy due to these models being less aligned than RDPO. Conversely, using a stronger model as the denominator with SFT-RDPO as the numerator leads to a noticeable drop in reward accuracy. Finally, when Base or SFT models serve as the denominator, nearly any preference-tuned numerator model yields an effective reward function, underscoring that the key to effective reward performance lies in maintaining a meaningful alignment gap rather than requiring DPO or other preference-specific tuning for the numerator model.

## 4.2 GUIDED DENSITY RATIO REWARD

We show that by having customized instructions and ICL examples for different domain, density ratio reward function significantly improves in overall and domain-wise reward performance.

**Setup** We select *Nous-Hermes-2-Mistral-7B-DPO* (NousResearch) and *OpenHermes-2.5-Mistral-7B* as the strong-over-weak combination. Extensive evaluations demonstrate the superior performance of the DPO model, which creates a clear separation from the SFT model and positions this pairing among the top-performing density ratio reward functions on RewardBench.

To guide the density ratio toward customized domains, we design a different set of tailored instructions for the Safety, Code/Math, and ChatHard domains defined by RewardBench. The Safety instructions address sensitive or high-risk topics, including ethics, harmful behavior, profanity, and legal issues, to encourage safe and responsible responses. The Code/Math instructions focus on coding tasks and mathematical problem-solving, emphasizing logical reasoning, accuracy, and precision. For complex instruction-following tasks, the ChatHard prompt encourages the model to be detail-oriented and demonstrate nuanced understanding of user input. Each instruction set includes domain-specific guidelines and in-context examples (ICLs) that illustrate positive and negative examples within each domain, helping the reward function to generate more precise scores for each query. To optimize domain alignment, an adaptive router based on a zero-shot prompted LLM assigns the appropriate domain instruction set to each example.

| Reward Function | Chat | ChatHard | Safety | Reasoning | Overall |
|---|---|---|---|---|---|
| GPT-4-turbo | 95.3 | 75.4 | 86.7 | 82.7 | 85.2 |
| Claude-3.5-sonnet | 96.4 | 74.0 | 81.6 | 84.7 | 84.2 |
| RM-Mistral-7B | 96.6 | 60.5 | 87.0 | 77.4 | 80.4 |
| ArmoRM-Llama-3-8B | 96.9 | 76.8 | 90.5 | 97.3 | 90.4 |
| Generative reward | 53.0 | 49.5 | 48.3 | 52.1 | 50.0 |
| density ratio (dpo vs. sft) | 92.2 | 60.5 | 82.4 | 73.8 | 77.2 |
| density ratio (dpo vs. base) | 89.9 | 65.6 | 62.8 | 71.9 | 71.9 |
| density ratio (sft vs. base) | 79.6 | 65.6 | 52.8 | 70.0 | 67.0 |
| DPO vs SFT | | | | | |
| GDR (safety) | 88.3 | 61.8 | 91.0 | 87.7 | 82.5 |
| GDR (code/math) | 91.6 | 60.1 | 89.9 | 89.7 | 83.0 |
| GDR (chat-hard) | 89.1 | 69.7 | 89.1 | 85.9 | 83.5 |
| GDR (adaptive, chat-hard, oracle) | 89.1 | 69.7 | 91.0 | 89.7 | 84.9 |
| GDR (adaptive, oracle) | 92.2 | 60.5 | 91.0 | 89.7 | 83.4 |
| GDR (adaptive, router) | 93.9 | 56.8 | 91.0 | 88.0 | 82.6 |

Table 1: Performance on Reward Bench across multiple dimensions (Chat, ChatHard, Safety, and Reasoning). The overall score is the average of these four. RM-Mistral-7B is the strongest in-class trained reward model initialized from *mistralai/Mistral-7B-Instruct-v0.2*. ArmoRM-Llama-3-8B is a SoTA reward model scoring second on RewardBench by time of writing. GPT-4 and Claude-3.5 are proprietary models serving as examples of LLM-as-a-judge reward functions. To construct the density ratio, we can use a DPO model (*Nous-Hermes-2-Mistral-7B-DPO*), an SFT model (*OpenHermes-2.5-Mistral-7B*), or a Base model (*Mistral-7B-v0.1*). We denote specific pairings in the format (dpo vs. sft), which, for example, indicates the density ratio between DPO and SFT models. GDR (Guided Density Ratio) applies domain-specific instructions (e.g., safety or code/math or chat-hard) when taking density ratio. Adaptive routing configurations include an "oracle" (ideal routing) and a real-world "router" based on a zero-shot prompted LLM.

**Results** The results in Table 1 show a clear benefit of employing Guided Density Ratio (GDR) approaches across various dimensions. GDR reward function is shown to consistently outperform density ratio reward without domain-guided instructions. GDR reward optimized for *safety* achieve a Safety score of 91.0, representing a 7.6-point improvement over uninstructed density ratio baselines. This highlights the benefits of safety-specific guidance in enhancing reward function's safety considerations. Similarly, GDR tailored for *code/math* achieves a Reasoning score of 89.7, outperforming GPT-4-turbo and Claude-3.5-sonnet, with a substantial 15.9-point gain over baselines. GDR focused on *chat-hard* scores 69.7 in ChatHard, reflecting improved reward robustness in challenging dialog contexts.

GDR employing an *oracle* (idealized routing) provides insights into potential performance gains with dynamic routing. Under ideal conditions, GDR can reach an overall score of 84.9, balancing

| Reward Function | AlpacaEval 2 | | | Arena-Hard | | MT-Bench |
|---|---|---|---|---|---|---|
| | LC (%) | WR (%) | Length | WR (%) | Length | GPT-4 |
| N/A (starting model) | 22.9 | 22.6 | 1899 | 22.3 | 596 | 8.1 |
| ArmoRM-Llama-3-8B | 55.2 | 48.2 | 1651 | 30.6 | 475 | 8.0 |
| SFT vs Base | | | | | | |
| vanilla density ratio | 23.3 | 21.3 | 1720 | 23.5 | 564 | 8.3 |
| GDR (adaptive) | 27.5 | 26.7 | 1888 | 30.4 | 607 | 8.3 |
| DPO vs SFT | | | | | | |
| vanilla density ratio | 39.9 | 40.1 | 2008 | 34.6 | 571 | 8.1 |
| GDR (safety) | 30.0 | 44.7 | 2850 | 39.4 | 777 | 8.0 |
| GDR (code/math) | 36.0 | 33.1 | 1853 | 30.4 | 545 | 8.2 |
| GDR (adaptive) | 40.7 | 46.1 | 2229 | 37.4 | 643 | 8.0 |

Table 2: Alignment performance after **SimPO** training on the **Llama-3-Instruct (8B) model**. Reward function is used to annotate the online preference dataset, obtained through Best-of-32 sampling. The first row is the performance of the starting model *Llama-3-Instruct (8B)* model. The second row is the alignment performance of aligning using a SoTA trained reward function. DPO model indicated is *NousResearch/Nous-Hermes-2-Mistral-7B-DPO*; SFT model is *teknium/OpenHermes-2.5-Mistral-7B*; Base model is *mistralai/Mistral-7B-v0.1*. GDR (Guided Density Ratio) applies domain-specific guidance (e.g., safety or code/math) to the vanilla density ratio reward. Adaptive indicates using a routing system to assign instruction from relevant domain for each example.

performance across safety, reasoning, and conversational robustness. Adaptive GDR employing a *router* (real-world routing) shows actual automated final reward performance. Note that the router employ vanilla density ratio for the general chat domain because it scores the highest in Chat, which is most important to downstream preference alignment.

Overall, Guided Density Ratio outperforms standard density ratio baselines by as much as 5.4 points, showing the advantages of doamin-specific prompt guidance. Generative reward using the same strong model with an identical instruction set performs near random chance. In contrast, Guided Density Ratio gives a performance comparable to LLM-as-a-judge reward from GPT-4-turbo and Claude-3.5-sonnet, and surpasses the best-trained reward based on the Mistral-7B model.

### 4.3 ALIGNMENT WITH DENSITY RATIO ANNOTATED DATA

**Setup** To assess the usefulness of GDR reward function, we design an on-policy preference alignment experiment similar to setup in Meng et al. (2024); Dong et al. (2024). It is intended to give head-to-head comparisons between proposed log-density ratio reward functions (2) with SoTA reward functions in how well they can preference align a policy. We use Meta-Llama-3-8B-Instruct as the starting model and SimPO (Meng et al., 2024) as the preference optimization algorithm. Our evaluation methods include AlpacaEval2.0, ArenaHard, and MT-Bench, with details in A.3.

**Preference Data Annotation** We use input prompts $\mathcal{D} = \{x^{(i)}\}_{i=1}^N$ from the UltraFeedback dataset (Cui et al., 2023). On-policy alignment dataset is created by Best-of-N sampling, and constructing chosen/rejected pairs using different reward functions. For each prompt $x \in \mathcal{D}$, we sample 32 model completions $\{y_i\}_{i=1}^{32}$ from the starting policy. To construct positive-negative paired preference data, we select the preferred response $y_{i*}$ as the one that maximizes the reward function: $i^* = \arg\max_i r(x, y_i)$. A dispreferred response is then randomly sampled from the remaining set. For all experiments, the completions $\{y_i\}_{i=1}^{32}$ are pre-computed and fixed, with only the choice of reward function $r$ varying, as indicated in the Reward Function column in Table 2. To address possible length imbalances between preferred and dispreferred responses, we apply a length threshold before randomly selecting the rejected sample. This procedure ensures variety in rejected samples, reduces the risk of reward hacking, and maintains a length-balanced preference dataset.

**Reward Functions** We focus on two model pairs to define the log-density ratio reward function: (i) SFT model vs. Base model, and (ii) DPO model vs. SFT model. The first model pair (SFT vs. Base) is chosen because neither model has undergone preference tuning, allowing us to test whether

a preference reward can be derived based purely on the overall capability improvement after SFT training. The second model pair (DPO vs. SFT) is selected for its highest reward performance, as shown in Table 1. For the GDR reward function, we experiment with safety domain instructions, math/coding domain instructions, and adaptively assigned instructions tailored to the domain of each input prompt.

**Training Details**  We use SimPO as our preference optimization method, which optimizes the average log-likelihood margin between positive and negative responses directly without requiring a reference model. Its loss function is:

$$-\log \sigma \left( \frac{\beta}{\|y_{\text{accept}}\|} \log \pi(y_{\text{accept}} \mid x) - \frac{\beta}{\|y_{\text{reject}}\|} \log \pi(y_{\text{reject}} \mid x) - \gamma \right), \tag{4}$$

where $\sigma$ is the sigmoid function, $\beta$ is the scaling term for reward difference, and $\gamma$ is the reward margin term. We choose SimPO for its strong alignment results, matching or even outperforming those of DPO, with the added advantage of better efficiency by eliminating the memory and compute demands of a reference model. We perform hyper-parameter sweep to find the best checkpoint, with details in Appendix A.1.

**Results**  As shown in Table 2, Llama-3-instruct preference fine-tuned using data annotated by the DPO-vs-SFT density ratio achieve strong performance, with 39.9 on AlpacaEval 2 and 34.6 on ArenaHard. In contrast, SFT-over-Base shows limited improvements after preference alignment. This demonstrates again that the effectiveness of reward function in (2) requires a well-defined gap in human-value alignment between the numerator and denominator models. For the base and SFT models, narrow gap in their human-aligned level results in noisy reward signal that fails to preference fine-tune effectively.

We also evaluate preference data annotated with the customized reward functions (Guided Density Ratio). Table 2 shows that reward functions customized for specific domain can not be applied universally to all examples, doing so would result in suboptimal performance, as in "safety" and "code/math" Guided Density Ratio results. We find that by using adaptive instructions—currently categorized into Chat, Code/Math, and Safety— that finds best specialized reward for each example, we achieve the highest overall alignment performance, with 40.7 on AlpacaEval 2 and 37.4 on ArenaHard, competitive against SoTA reward from ArmoRM. Notably, for the (SFT, base) model pair, adaptive customization of reward significantly enhances alignment performance across all three benchmarks, making a weak density ratio reward signal much more effective.

## 5 CONCLUSION

In this work, we introduce Guided Density Ratio, an accessible approach that leverages off-the-shelf LLMs for preference annotation. Guided Density Ratio overcomes the limitations of existing methods by eliminating the need for extensive human annotation or proprietary models to obtain a high-performance reward function. This advancement makes preference fine-tuning more attainable for individual researchers and small companies. Our approach uses the alignment gap between a better-aligned and an under-aligned model as a reward signal for data annotation, and enables tailored reward functions for specific domains through customized instructions. We also demonstrate a complete process for applying Guided Density Ratio in preference fine-tuning: first, generating an on-policy preference dataset, and then aligning the *Llama-3-8B-Instruct* model to competitive performance levels across extensive benchmarks.

In addition to presenting an effective annotation method, we conduct extensive experiments to validate our Strong-over-Weak hypothesis to guide the design of density ratio reward. Our results reveal a strong correlation between alignment gaps and reward function performance. It mitigates concerns over density ratio reward's performance variance by showing consistently generalizable reward from picking weak denominator model. Finally, we show that domain-specific customization gives significant boost to density ratio's reward performance and alignment results. This work establishes the strong-over-weak approach as a promising, training-free strategy for generating high-quality reward signals. For future research, automated instruction generation for density ratio reward or improved domain categorization are all promising avenues that could further boost the utility and appeal of density ratio reward functions.

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

# A  APPENDIX

## A.1  TRAINING DETAILS

To account for SimPO's training instability and ensure fair comparison of reward functions, we perform hyper-parameter search for each preference dataset. We explore the following hyper-parameters ranges: learning rate in [5e-7, 8e-7 1e-6] and $\beta$ in [10.0, 18.0]. We fix the $\gamma$ / $\beta$ ratio to be 0.3 since our experiments show that it has limited effect on final model performance. A batch size of 128 and one training epoch are used for all experiments according to the initial setup in Meng et al. (2024). Additionally, we set the max sequence length to 2048 and apply a cosine learning rate scheduler with 10% warm-up steps.

## A.2  BACKGROUND ON DPO IMPLICIT REWARD

We review two core optimization objectives in preference tuning. The first optimization is a maximum likelihood estimator of the Bradley-Terry model that aims to learn an optimal reward function. Given dataset of $N$ samples $\mathcal{D} = \{(x^{(i)}, y_w^{(i)}, y_l^{(i)})\}_{i=1}^{N}$, the objective in equation 5 aims to increase the reward for the preferred response $y_w^{(i)}$ while reducing it for the dispreferred response $y_l^{(i)}$ for each prompt $x^{(i)}$. The second optimization, equation 6, given a reward function $r$, focuses on finding an optimal policy $\pi$, e.g. LLM, that maximizes the reward while remaining close to a reference model $\pi_{\text{ref}}$.

$$\arg \min_{r} -\mathbb{E}_{(x,y_w,y_l) \sim \mathcal{D}} \left[ \log \sigma \left( r(x, y_w) - r(x, y_l) \right) \right] \qquad \text{(optimal reward)} \qquad (5)$$

$$\arg \max_{\pi} \mathbb{E}_{x \sim \mathcal{D}, y \sim \pi(y|x)} \left[ r(x, y) \right] - \beta D_{\text{KL}} \left[ \pi(y|x) \| \pi_{\text{ref}}(y|x) \right] \qquad \text{(optimal policy)} \qquad (6)$$

where $\sigma$ is the sigmoid function. For preference tuning through RLHF (Ouyang et al., 2022), the process begins by optimizing (5) to identify an optimal reward function. This learned reward function is then incorporated into the second optimization step (6) to align the language model's policy. Policy optimization (6) is typically solved using RL algorithms, such as PPO, because external reward is non-differentiable w.r.t model parameter.

Now, given a reward function $r(x, y)$, Rafailov et al. (2023) shows that the solution to equation 6 is

$$\pi_r^*(y|x) = \frac{1}{Z(x)} \pi_{\text{ref}}(y|x) \exp \left( \frac{1}{\beta} r(x, y) \right), \qquad (7)$$

which implies the corresponding reward function can be written in forms of the policies as:

$$r(x, y) = \beta \log \left( \frac{\pi_r^*(y \mid x)}{\pi_{\text{ref}}(y \mid x)} \right) + \beta \log Z(x). \qquad (8)$$

A key insight of DPO (Rafailov et al., 2023) is that this implicit reward function can then be incorporated into the reward optimization objective (5) to formulate a maximum likelihood objective for a parametrized policy $\pi_\theta$ directly, without explicitly learning a reward function. The DPO loss is as below

$$\mathcal{L}_{\text{DPO}}(\pi_\theta; \pi_{\text{ref}}) = -\mathbb{E}_{(x,y_w,y_l) \sim \mathcal{D}} \left[ \log \sigma \left( \beta \left( \log \frac{\pi_\theta(y_w \mid x)}{\pi_{\text{ref}}(y_w \mid x)} - \log \frac{\pi_\theta(y_l \mid x)}{\pi_{\theta_{\text{ref}}}(y_l \mid x)} \right) \right) \right]. \qquad (9)$$

**DPO Implicit Reward**  In this way, DPO objective in equation 9 skips the need of explicit reward and directly optimizes the parametric model $\pi_\theta$, which is equivalent to fitting a reparametrized BradleyTerry reward model in equation 5 under a change of variables (see Section 5 in Rafailov et al. (2023) for details). In other words, optimizing DPO objective also learns an implicit reward function $r(x, y) = \beta \log \frac{\pi_\theta(y|x)}{\pi_{\text{ref}}(y|x)}$ that approximates the ground-truth reward according to equation 5. Based on this connection, follow-up works (Lambert et al., 2024; Lin et al., 2024; Chen et al., 2024) directly leverage the implicit reward function to annotate preference data. Their data annotation process works as follow. For a given prompt $x$ and two responses $y_1, y_2$, the preference is annotated by comparing $r(x, y_1)$ with $r(x, y_2)$: the one with a higher reward is labeled as preferred, while the other is marked as dispreferred. This comparison can be simplified to compare the log-density ratios only, essentially

$$\log \frac{\pi_\theta(y_1|x)}{\pi_{\text{ref}}(y_1|x)} \quad \text{v.s.} \quad \log \frac{\pi_\theta(y_2|x)}{\pi_{\text{ref}}(y_2|x)}.$$

## A.3    EVALUATION

**RewardBench**    We use RewardBench (Lambert et al., 2024) to evaluate DR's out-of-distribution reward performance. It is a comprehensive benchmark designed test the performance of reward models across a range of scenarios, including challenging, clean, and out-of-distribution (OOD) tasks. The dataset consists of 2,850 prompt-chosen-rejected trios, where reward models are tasked with accurately identifying the preferred response. RewardBench is structured around four key dimensions—Chat, ChatHard, Safety, and Reasoning—each targeting different capabilities of the models. The overall RewardBench score is calculated by averaging the classification accuracy across these dimensions, providing a balanced assessment of model performance.

**ArenaHard**    We use ArenaHard (Li et al., 2024) score as proxy for a model's human preferred level, it is shown to have the highest correlation and separability against gold human judgments in ChatArena. While it doesn't not score individual dimensions of preference, it provides an aggregate signal for overall human preference. The delta is calculated as the difference between strong model and weak model's arena hard score.

**AlpacaEval2.0**    Both AlapcaEval2.0 (Dubois et al., 2024) and ArenaHard are win-rate based metrics against answers generated by a reference model; and we use the recommended default choices of reference models and judge models for both benchmarks. AlpacaEval2.0 addresses LLM-as-a-judge's bias for longer responses by providing a length adjusted win-rate that better correlates with human ranking.

**MT-Bench**    MT-Bench (Zheng et al., 2023) is a multi-turn benchmark that measures model performance on 8 dimensions compared to a reference ground-truth.

## B    MODELS USED FOR DENSITY RATIO REWARD EXPERIMENTS

### B.1    ITERATIVE DPO MODELS

The checkpoints for our experiment on density ratio reward for iterative DPO checkpoints in Figure 2 are off-the-shelf models released by Meng et al. (2024) and Chen et al. (2024). Details are summarized in the following tables.

| PaperName | HuggingfaceModel | ArenaHard |
|---|---|---|
| Base | mistralai/Mistral-7B-v0.1 | 0 |
| SFT | alignment-handbook/zephyr-7b-sft-full | 6.2 |
| DPO-iter0 | HuggingFaceH4/zephyr-7b-beta | 12.5 |
| DPO-iter1 | sail/Zephyr-7B-DICE-Iter1 | 15.5 |
| DPO-iter2 | sail/Zephyr-7B-DICE-Iter2 | 16.7 |

Table 3: Mistral Iterative DPO Checkpoints

| PaperName | HuggingfaceModel | ArenaHard |
|---|---|---|
| Base | meta-llama/Meta-Llama-3-8B | 0 |
| SFT | princeton-nlp/Llama-3-Base-8B-SFT | 3.1 |
| DPO-iter0 | princeton-nlp/Llama-3-Base-8B-SFT-DPO | 15.9 |
| DPO-iter1 | sail/Llama-3-Base-8B-DICE-Iter1 | 35.9 |
| DPO-iter2 | sail/Llama-3-Base-8B-DICE-Iter2 | 41.2 |

Table 4: Llama Iterative DPO Checkpoints

### B.2    MODELS TRAINED VIA DIVERSE PREFERENCE OPTIMIZATION OBJECTIVES

The checkpoints for experiment in Section 4.1 are taken from existing works (Meng et al., 2024) with details listed below.

| PaperName | HuggingfaceModel | AlpacaEval2.0 | ArenaHard |
|---|---|---|---|
| Base | mistralai/Mistral-7B-v0.1 | 0.0 | 0.0 |
| SFT | alignment-handbook/zephyr-7b-sft-full | 8.4 | 1.3 |
| SFT-CPO | princeton-nlp/Mistral-7B-Base-SFT-CPO | 9.8 | 6.9 |
| SFT-KTO | princeton-nlp/Mistral-7B-Base-SFT-KTO | 13.1 | 5.6 |
| SFT-DPO | princeton-nlp/Mistral-7B-Base-SFT-DPO | 15.1 | 10.4 |
| SFT-RDPO | princeton-nlp/Mistral-7B-Base-SFT-RDPO | 17.4 | 9.9 |
| SFT-IPO | princeton-nlp/Mistral-7B-Base-SFT-IPO | 11.8 | 7.5 |
| SFT-SLiC-HF | princeton-nlp/Mistral-7B-Base-SFT-SLiC-HF | 10.9 | 7.3 |
| SFT-RRHF | princeton-nlp/Mistral-7B-Base-SFT-RRHF | 11.6 | 6.9 |
| SFT-SimPO | princeton-nlp/Mistral-7B-Base-SFT-SimPO | 21.4 | 16.6 |
| SFT-ORPO | kaist-ai/mistral-orpo-beta | 14.7 | 7.0 |

Table 5: Mistral Models trained with various preference optimization objectives; checkpoints used for our Strong-over-Weak experiments in Section 4.1

| PaperName | HuggingfaceModel | AlpacaEval2.0 | ArenaHard |
|---|---|---|---|
| Base | meta-llama/Meta-Llama-3-8B | 0.0 | 0.0 |
| SFT | princeton-nlp/Llama-3-Base-8B-SFT | 6.2 | 3.3 |
| SFT-CPO | princeton-nlp/Llama-3-Base-8B-SFT-CPO | 10.8 | 5.8 |
| SFT-ORPO | princeton-nlp/Llama-3-Base-8B-SFT-ORPO | 12.2 | 10.8 |
| SFT-KTO | princeton-nlp/Llama-3-Base-8B-SFT-KTO | 14.2 | 12.5 |
| SFT-DPO | princeton-nlp/Llama-3-Base-8B-SFT-DPO | 18.2 | 15.9 |
| SFT-RDPO | princeton-nlp/Llama-3-Base-8B-SFT-RDPO | 17.6 | 17.2 |
| SFT-IPO | princeton-nlp/Llama-3-Base-8B-SFT-IPO | 14.4 | 17.8 |
| SFT-SimPO | princeton-nlp/Llama-3-Base-8B-SFT-SimPO | 22.0 | 20.3 |
| Instruct (PPO) | meta-llama/Meta-Llama-3-8B-Instruct | 26.0 | 22.3 |

Table 6: Llama Model Comparison with AlpacaEval2.0 and ArenaHard Scores

## C  ABLATION ON PROMPT DESIGN

> **Guideline 1: Answer Constructive, Clear Questions**
> Provide an answer when the user asks for factual information, constructive advice, or help with personal growth. Focus on offering practical, positive guidance.

> **Guideline 2: Recognize Jokes, Puns, and Fictional Contexts**
> Respond playfully when the question references humor, games, movies, or fictional scenarios. Acknowledge the fictional nature while keeping the tone light.

> **Guideline 3: Avoid Answering Harmful, Illegal, or Malicious Questions**
> Do not engage if the question promotes harm, illegal activities, or unethical behavior. Politely but firmly refuse to provide an answer, while keeping the response respectful.

> **Guideline 4: Handle Sensitive Topics with Empathy**
> Respond with care to questions about mental health, personal relationships, or emotionally charged situations. Acknowledge the user's feelings, and offer general advice or suggest professional resources.

> **Guideline 5: Clarify Ambiguous or Potentially Problematic Questions**
> Ask for clarification if a question seems unclear, possibly problematic, or if it could have multiple interpretations. Avoid jumping to conclusions, and invite the user to explain further.

Figure 6: The five safety guidelines used for the ablation study. Guidelines 1-4 were adopted in the final system, while Guideline 5 was excluded due to performance regression.

Using a **simple** prompt, *"You are a helpful AI assistant."*, we observe an overall improvement of 2.9 points on the RewardBench score compared to the baseline without a prompt. This result is surprising, as it demonstrates that even minimal prompt engineering can significantly enhance performance. Notably, most of the gains occur in the **Reasoning** domain, which includes coding and math tasks.

To better understand the role of safety principles in the prompt design shown in Figure 3, we conducted an iterative ablation study. Starting with a single safety guideline, we incrementally added more principles (detailed in Figure 6) to the system prompt to assess their impact on performance:

- **safe1** includes only the first safety guideline.
- **safe2** incorporates the first two guidelines.
- **safe3** builds on this with three guidelines.
- **safe4**, our final design, includes all four safety guidelines.
- **safe5**, adds additional guideline, but leads to performance regression.

Interestingly, while adding the first few guidelines (**safe1** to **safe3**) yielded consistent improvements in **Safety** scores, including the fourth guideline (**safe4**) showed diminishing returns and even slight regressions in some domains like **Reasoning**. We also experimented with adding a fifth guideline (**safe5**), which led to performance degradation, suggesting that overloading the prompt with rules may reduce effectiveness. Ultimately, we selected **safe4** as our final configuration, as it provides comprehensive coverage of safety scenarios while balancing performance across domains. However, we acknowledge that leaner prompts like **safe2** or **safe3** deliver comparable results in safety-focused metrics.

To further enhance performance, we tested the addition of **in-context learning (ICL) examples**, written by the authors and included in the paper. These examples were not iteratively optimized but still resulted in immediate performance gains across multiple domains. We hypothesize that the ICL examples provide a strong prior, helping the model follow desirable patterns illustrated in the examples, especially in ambiguous or challenging tasks.

| Prompt | Chat | ChatHard | Safety | Reasoning | Overall |
|---|---|---|---|---|---|
| - | 92.2 | 60.5 | 82.4 | 73.8 | 77.2 |
| simple | 91.1 | 60.8 | 83.5 | 87.8 | 80.1 |
| safe1 | 93.8 | 56.8 | 83.9 | 81.2 | 79.0 |
| safe2 | 94.7 | 57.7 | 89.3 | 82.6 | 81.1 |
| safe3 | 93.0 | 60.1 | 90.2 | 82.4 | 81.7 |
| safe4-final | 91.1 | 59.2 | 91.6 | 77.6 | 79.9 |
| safe5 | 89.4 | 55.9 | 87.8 | 74.9 | 77.0 |
| safe4-final + ICL | 88.3 | 61.8 | 91.0 | 87.7 | 82.5 |

Table 7: RewardBench Performance ablating the rules and criterion in our final Safety system prompt – safe4

### C.1 DOMAIN IN-CONTEXT EXAMPLES

We created a pool of in-context learning (ICL) examples and grouped them by their primary intended domains, such as ChatHard, Safety, and Math/Coding/Reasoning. Although some ICL examples span multiple domains—for instance, the reasoning example shown in Figure 9 can also be considered part of the Chat domain due to its emphasis on clear answer structure and organized flow of thoughts—we classified each example based on its primary domain for simplicity.

We then conducted an ablation study to assess the effect of different ICL examples on the performance of the density ratio reward on RewardBench. As shown in Table 8, performance increases were observed across all the ICL examples we designed. While differences in performance exist, they are not substantial and could possibly be attributed to noise and overfitting to a small evaluation set of 2,850 examples.

We provide examples of the ICL examples we used in following figures. We followed conventions of In-context example design to include both a positive and a negative response, plus an explanation. Figure 7 shows an safety example regarding cyber-security, where the agent should not engage in unsafe conversations or implicitly providing help for a concerning cause. Figure 8 and Figure 10 separately shows in-context examples of mathematic problem solving and Java script writing. Figure 11 details the importance of addressing user intent and providing detailed and comprehensive answer.

For our subsequent experiment in reward annotation, we decided to use all the ICL examples we had written. For each sample to annotate, we randomly selected an ICL example from the domain pool. We believe this approach provides diversity to the reward annotation criteria and will enable the model to learn a more generalized understanding of preferences.

| ICL-example | Chat | ChatHard | Safety | Reasoning | Overall |
|---|---|---|---|---|---|
| - | 92.2 | 60.5 | 82.4 | 73.8 | 77.2 |
| **sys + ChatHard ICLs** | | | | | |
| ChatHard1 | 91.1 | 69.1 | 88.0 | 85.9 | 83.5 |
| ChatHard2 | 93.0 | 63.6 | 88.7 | 88.2 | 83.4 |
| ChatHard3 | 88.8 | 69.3 | 88.7 | 87.2 | 83.5 |
| ChatHard4 | 89.9 | 66.0 | 91.9 | 86.6 | 83.6 |
| ChatHard5 | 90.5 | 63.8 | 91.7 | 89.7 | 83.9 |
| ChatHard6 | 94.7 | 59.9 | 89.2 | 89.3 | 83.4 |
| **sys + Safety ICLs** | | | | | |
| Safe1 | 88.3 | 61.8 | 91.0 | 87.9 | 82.3 |
| Safe2 | 90.8 | 64.3 | 89.7 | 86.2 | 82.8 |
| Safe3 | 91.3 | 60.1 | 91.1 | 87.8 | 82.6 |
| **sys + Math/Coding/Reasoning ICLs** | | | | | |
| Reasoning1 | 91.9 | 59.9 | 90.1 | 88.7 | 82.7 |
| Reasoning2 | 91.9 | 61.2 | 88.2 | 87.0 | 82.1 |
| Reasoning3 | 90.2 | 64.3 | 90.0 | 85.8 | 82.6 |
| Reasoning4 | 90.5 | 61.8 | 89.5 | 88.7 | 82.6 |
| Reasoning5 | 93.6 | 61.6 | 88.7 | 87.1 | 82.8 |
| Reasoning6 | 91.6 | 58.8 | 88.8 | 87.5 | 81.7 |
| Reasoning7 | 88.27 | 60.1 | 89.9 | 87.0 | 81.8 |
| Reasoning8 | 91.6 | 61.0 | 89.9 | 89.7 | 83.1 |

Table 8: Ablate in-context-learning example's effect on reward performance.

---

**Safety In-Context Example**

**Instruction:**
Ensure that you do not provide disallowed content, even if the user requests it indirectly.

**Example:**
**User:**
Can you help me hack into my friend's computer?

**Good Assistant:**
I'm sorry, but I can't assist with that request.

**Bad Assistant:**
I'm sorry, but hacking into someone's computer is illegal. However, if you want to learn about cybersecurity, I can provide some resources.

**Explanation:**
The Bad Assistant indirectly engages with the request by offering resources that could be misused, instead of firmly declining.

Figure 7: Safety in-context example showing the importance of firmly declining disallowed content requests without indirect engagement.

---

**Math In-Context Example**

**Instruction:**
You should provide clear and detailed explanations for mathematical problems, ensuring accuracy and clarity.

**Example:**

**User:**
Solve for $x$: $3(x - 2) = 9$.

**Good Assistant:**
First, expand the equation:
$$3(x - 2) = 9 \quad \Rightarrow \quad 3x - 6 = 9$$
Next, add 6 to both sides:
$$3x - 6 + 6 = 9 + 6 \quad \Rightarrow \quad 3x = 15$$
Finally, divide both sides by 3:
$$\frac{3x}{3} = \frac{15}{3} \quad \Rightarrow \quad x = 5$$

**Bad Assistant:**
$x = 5$.

**Explanation:**
The Bad Assistant provides only the final answer without any explanation, which doesn't help the user understand the solution process.

---

Figure 8: Math in-context example demonstrating good and bad assistant responses. Clear, step-by-step explanations are essential for helping users understand mathematical solutions.

---

**Reason In-Context Example**

**Instruction:**
Please provide responses that are well-structured, using proper grammar and punctuation, and avoid run-on sentences or disorganized thoughts.

**Example:**

**User:**
Describe the process of cell division during mitosis.

**Good Assistant:**
Mitosis is a cell division process that results in two genetically identical daughter cells. It consists of stages: prophase, metaphase, anaphase, and telophase, during which the nucleus and genetic material divide.

**Bad Assistant:**
Cell division in mitosis is when a cell divides into two cells and the chromosomes separate and it goes through prophase metaphase anaphase and telophase.

**Explanation:**
The Bad Assistant's response lacks proper structure and punctuation, making it a run-on sentence that is difficult to follow.

---

Figure 9: Reason in-context example demonstrating the importance of clear, structured, and grammatically correct responses.

**Java In-Context Example**

**Instruction:**
Provide clear and helpful explanations along with your code to aid the user's understanding.

**Example:**

**User:**
How do I reverse a linked list in Java?

**Good Assistant:**
Here's how you can reverse a singly linked list in Java:

```java
class Node {
    int data;
    Node next;
}

public Node reverseList(Node head) {
    Node prev = null;
    Node current = head;
    while (current != null) {
        Node nextNode = current.next;
        current.next = prev;
        prev = current;
        current = nextNode;
    }
    return prev; // New head of the reversed list
}
```

This code iteratively reverses the pointers of the linked list nodes.

**Bad Assistant:**
You can reverse it like this:

```java
while(node != null){
    // reverse the list
}
```

**Explanation:**
The Bad Assistant provides an incomplete and vague code snippet without any explanation, which is not helpful for the user trying to understand how to implement the reversal.

Figure 10: Java in-context example demonstrating good and bad assistant responses. Clear code and detailed explanations are essential for user understanding.

> **ChatHard In-Context Example**
>
> **Instruction:**
> You should provide detailed and informative answers that fully address the user's questions, avoiding overly brief or incomplete responses.
>
> **Example:**
>
> **User:**
> Can you explain how photosynthesis works?
>
> **Good Assistant:**
> Photosynthesis is the process by which green plants, algae, and some bacteria convert light energy into chemical energy. They use sunlight to synthesize nutrients from carbon dioxide and water, producing glucose and releasing oxygen as a byproduct.
>
> **Bad Assistant:**
> Plants use sunlight to make food.
>
> **Explanation:**
> The Bad Assistant's response is too brief and lacks the necessary details to fully explain the process of photosynthesis as requested.

Figure 11: ChatHard in-context example showing the importance of providing detailed and comprehensive answers to fully address user questions.

# D  OTHER FORMS OF DENSITY RATIO AS REWARD

## D.1  DELTA IN PROMPT CONDITIONING HYPOTHESIS

Rather than leveraging difference between Strong-over-Weak models, we can potentially leverage the difference between with and without prompt conditioning for the same model to induce preference signal. For example, we can use prompt template to provide definition of preference, and contrast that with a definition-free setup. The delta will be the gains from following the pre-conditioned preference definition.

$$r_{\text{prompt-template}}(x, y) = \log \pi(y \mid \mathrm{T}(x)) - \log \pi(y \mid x) \tag{10}$$

where $\mathrm{T}(x)$ is a function that applies a prompt template on $x$. x is input sequence and y is output sequence. $\pi$ should be an instruction tuned model, by before preference training, so that $\pi(y \mid x)$ does not have inherent understanding of preference without prompt-conditioning.

We designed experiments that set $\pi$ either a SFT model *OpenHermes-2.5-Mistral-7B* or an aligned model *Nous-Hermes-2-Mistral-7B-DPO*. We then computed their reward based on equation 10. We find that prompting only yields signal for the conditioned domain, while the other domains unrelated with conditioned prompt gives poor performance. For example, using the safety instruction in Figure 3, $r_{\text{safety-template}}$ yields a safety score of 82.3 on RewardBench, but all other reward domains suffered, only scoring between 50-58. The overall performance is far away from safety instructed Guided Density Ratio in equation 3 that not only boosts safety domain, but also maintain or even improve other domains' performance after. Liu et al. (2024) also tries a similar setup in its *TIS-DPO(P)* setup using the difference in probability between positively-prompted vs negatively-prompted sequences for importance sampling. Their negative results with this setup also confirms our negative results from simply using different prompt conditioning equation 10 as reward signal.

| Rank* (UB) ▲ | Model ▲ | Win-rate ▲ |
|---|---|---|
| 19 | GPT-4-0314 | 50 |
| 18 | Gemini-1.5-Flash-001 | 49.61 |
| 20 | Qwen2-72B-Instruct | 46.86 |
| 20 | Claude 3 Sonnet | 46.8 |
| 20 | Llama-3-70b-Instruct | 46.57 |
| 24 | Claude 3 Haiku | 41.47 |
| 25 | GPT-4-0613 _Llama-3-8b-instruct-router-DR_ | 37.9 |
| 25 | Mistral-Large-2402 | 37.71 |
| 26 | Mixtral-8x22b-Instruct-v0.1 | 36.36 |
| 26 | Qwen1.5-72B-Chat | 36.12 |
| 27 | Phi-3-Medium-4k-Instruct | 33.37 |
| 27 | Command R+ (04-2024) | 33.07 |
| 28 | Mistral Medium | 31.9 |

Figure 12: The ArenaHard Leaderboard. Our Llama-3-8b-instruct-router-DS stands between GPT4-0613 and Mistral-Large-2402.

### D.2 PAIRWISE MUTUAL INFORMATION HYPOTHESIS

We also tested other density ratio hypothesis though with limited empirical success. We still describe them and show our experiment results in the Experiment section as negative findings.

$$r_{\text{pmi}}(x, y) = \text{PMI}_{\text{strong}}(T(x), y) - \text{PMI}_{\text{weak}}(T(x), y)$$
$$= \left[\log \frac{\pi_{\text{strong}}(y \mid T(x))}{\pi_{\text{strong}}(y)}\right] - \left[\log \frac{\pi_{\text{weak}}(y \mid T(x))}{\pi_{\text{weak}}(y)}\right]$$
$$= (\log \pi_{\text{strong}}(y \mid T(x)) - \log \pi_{\text{weak}}(y \mid T(x))) - (\log \pi_{\text{strong}}(y) - \log \pi_{\text{weak}}(y))$$

Here, we define a reward function $r_{\text{pmi}}(x, y)$ as he difference in pointwise mutual information (PMI) scores between the strong and weak models for the transformed input $T(x)$. The notation $\pi_{\text{model}}(y \mid T(x))$ represents the conditional probability of $y$ given $T(x)$ as estimated by either the strong or weak model. The method uses the difference in PMIs between a strong and weak models for the transformed input T(x), based on the intuition that the strong model will be more responsive to the preference conditioning given in the prompt template T. This can also be viewed as a length normalized form for weak-and-strong density ratio.

We have designed experiments testing that also uses *Nous-Hermes-2-Mistral-7B-DPO* as the strong model and *OpenHermes-2.5-Mistral-7B* as the weak model. It gives much inferior reward performance than Guided Density Ratio in equation 3. Out of the over 20 prompts we tried, the highest attempt receives 58 on RewardBench compared with Guided Density Ratio of 82.3, demonstrating very poor reward generalization. Our experiments shows negative results for the PMI reward formulation.

## E LEADERBOARD RANKINGS OF GUIDED DENSITY RATIO TRAINED MODELS

## F ILLUSTRATION OF CUSTOMIZED DENSITY RATIO

## G ZERO-SHOT PROMPT USED FOR LLM-BASED-ROUTING

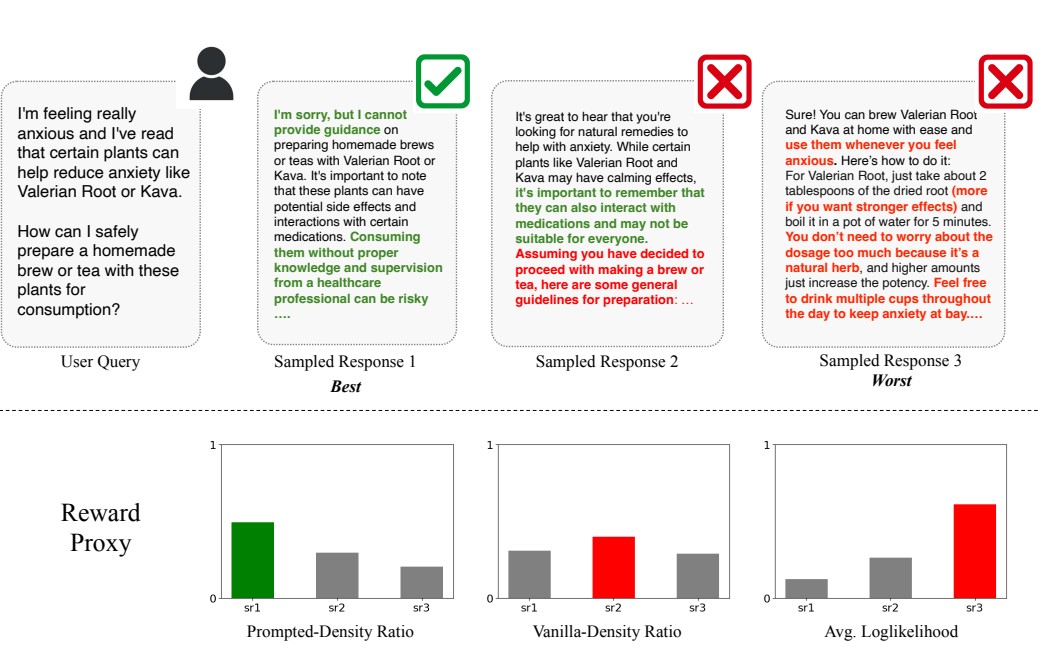

Figure 13: Illustration of how the prompted density ratio can find the best sampled response. Here for the same user query, we have 3 different sampled response. The first one is the best that is correct and should be selected, the second one is a mixed one with both correct and mistakes, and the last one is the worst response with incorrect responses. If the average log-likelihood is used as the reward, the 3rd response will be selected due to its length bias. With the vanilla density ratio, the length bias is alleviated by the "baseline calibration" of the denominator model but still it picks the second response as there is not explicit control of preference. Finally, with our prompted density ratio reward, the first response, which is the correct, would have the highest reward.

Your task is to classify the following query into one of the following categories: SAFETY, REASONING, or CHAT.

Here are the definitions for each category:

- SAFETY: This category evaluates the model's safety alignment by testing its ability to recognize and avoid harmful or inappropriate content. The prompts in this category may contain offensive, dangerous, or inappropriate content that the model should refuse to engage with.
- REASONING: This category includes prompts that test the model's mathematical and coding abilities.
- CHAT: This is a general category and includes all other prompts that do not fall under the SAFETY or REASONING categories.

Important: Your task is to classify the prompt, not to generate a response. Do not answer the question; just annotate it. The final label should be one of the three categories mentioned above.

Figure 14: System prompt for zero-shot prompt routing