# OpenReview forum: "Preference Data Annotation with Guided Density Ratios"
_ICLR.cc/2025/Conference — Submitted to ICLR 2025_

### Official Review · Reviewer_ojiV · 2024-11-03

**Soundness:** 3
**Presentation:** 3
**Contribution:** 2
**Rating:** 5
**Confidence:** 3

**Summary:**

The paper investigates the preference tunning in LLM and proposes the challenge that it depends on paired human feedback data and the collection is costly in time and resources. Then the authors propose a novel data annotation technique that improves the density ratio rewards annotation accuracy and enhances the effectiveness of preference tuning using a new annotated dataset.

**Strengths:**

+ The paper explores the challenge of data scarcity in preference tuning, a significant question in the LLM era.

+ The author proposes a novel data annotation technique based on the prompt density ratio, which has advanced community skills and tunning-free advantages.

+ Extensive experiments include multiple settings on the cutting-edged open-source and close-resource MLLM to improve the solidity of the paper.

**Weaknesses:**

- While the proposed prompt-based density ratio method exhibits some improvements, the design of prompt templates for system prompts and few-shot prompts is widely used in broad scenarios with less innovation.

- Prompt-density ratio compared to vanilla-density ratio decreases the model’s bias, but if the model itself with some bias, it will accumulate to the updated modal, it should expand some discussion about the difference of bias between source and target model.

- The experiment section should expand a paragraph to summarise the setting and the experiment description is unclear. While I was confused by some of the questions listed below.

What is the generative reward in Table 1?

Why the second to last only include safety and reasoning?

Why are the results of NousResearch/Nous-Hermes-2-Mistral-7B-DPO not included in the Table, which is also an instruction-tuned model?

Why is only one setting included with a base or without a base that corresponds to two different weak models?

Does the experiment include a domain-specific prompt that should also include the comparison?

- There is a lack of ablation studies on the functionality of system prompts and few-shot prompts.

- Some sentences contain long-tail expressions, and several equations need improvement, particularly by adding appropriate punctuation marks.

**Questions:**

- How to ensure the annotate quality in the prompt routing for category-specific prompts and preference tunning data creation?

---

> ### Author Response · Authors · 2024-11-20
> **Response to Reviewer Comments**
>
> We thank the reviewer for their insight and constructive feedback. Below are responses to the reviewer’s specific questions.
>
> ---
>
> ### **W1:**
> We address the novelty concern of prompting in three aspects:
>
> 1. **Beyond generative tasks**: While system prompts and ICL examples are commonly applied to *generative tasks*, our work is the first to study their effects on density ratio reward models. Previous works (Lambert et al., 2024; Lin et al., 2024;) have used density ratio rewards, but none have employed instructions or prompts to guide or customize the reward function. To our knowledge, prompting has not been widely used to evaluate the likelihood of samples in general.
>
> 2. **Counter-intuitive to the DPO theory** (see **Common Response**): DPO implicit reward (density ratio between DPO-ed and pre-DPO models) is considered to directly approximate the ground truth reward distribution on *training data*. Applying prompts or instructions to an optimized reward model is counter-intuitive, and is thus not adopted by other works. We propose a different perspective to density ratio through **Strong-over-Weak Hypothesis**, which allows for prompting to improve controllability.
>
> 3. **Rules and principles as prompts in preference annotation**: The ability to craft rules and principles to supervise AI models is indeed critical for safe and controllable AI development. Prompting is a cost-effective way to encode “human preferences” that can vary across user groups and evolve over time. Our training-free approach offers significant advantages over static classifiers, which must be retrained with updated data. A concurrent work from OpenAI(Mu et al., 2024) also addresses the issue by proposing rule-based prompting for generative safety classifiers.
>
> ---
>
> ### **W2:**
> We appreciate the comment but may not have fully understood reviewer question. Could the reviewer clarify what is meant by the "updated model," the “source” model, and the “target” model?
>
> If the question pertains to the choice of models in the density ratio formulation, **Response-to-All-Reviewers (Part 1)** extensively discusses our updates and findings on the topic.
>
> ---
>
> ### **W3 (Experiment Design):**
> We revise the experiment design section (Section 4.2) to clarify the setting. Below are answers to specific questions:
>
> > **Q1:**
> Generative Reward uses the `Nous-Hermes-2-Mistral-7B-DPO` model to generate preferred or dispreferred judgments. The same system instruction, ICL examples, and routing mechanism were used for the generative baseline. But it still reports poor performance (equivalent to random guessing). Generative reward typically needs very strong models to be reliable.
>
> > **Q2:**
> We initially excluded results using chat-hard prompt template to avoid confusing readers, since this setup is not eventually adopted in our final method. We include the result for this setup in the updated Table-1.
> Chat and ChatHard are essentially the same category of samples, so we merge them into general Chat category. The templates for ChatHard lowered performance on the "simple" Chat domain, raising concern over its generalization. Thus, we eventually did not adopt it for our final method.
>
> > **Q3:**
> Results for `Nous-Hermes-2-Mistral-7B-DPO` are included both as density ratio rewards in the DPO vs. SFT sections of Table 1 and as Generative Reward. We have updated Table 1 for clarity.
>
> > **Q4:**
> We updated the table to better explain our model choice. Fixing the strong model as `Nous-Hermes-2-Mistral-7B-DPO`, we experiment with using Base or SFT as the weak model. Results show that using SFT as denominator yields better performance, so follow-up experiments mainly use the DPO vs. SFT model pair.
> We also test SFT vs. Base (neither trained on preferences), demonstrating that the density ratio between them can function as a reward signal. Surprisingly, this approach successfully aligns the `Llama-3-8B` model after applying prompt guidance, as shown in line 439 of Table 2.
>
> > **Q5 & W4:**
> We add detailed ablation studies on prompts and ICLs in Appendix C. Takeaways and findings from these ablations are available in **Response-to-All-Reviewers (Parts 2, 3)**.
>
> ---
>
> ### **Q1:**
> The router in our pipeline is a classifier, which can be evaluated using standard classification performance metrics. It is implemented using zero-shot prompting (Figure 14) on `Mixtral-8x7B-Instruct-0.1`, achieving 83% accuracy in 3-way classification (Chat, Safety, Reasoning).
> Despite its imperfections, the router has limited negative impact on performance, as shown in Table 1: the score with a perfect router is 83.4, compared to 82.6 with the zero-shot router.
>
> ---
> ### **References:**
>
> Rule-Based Rewards for Language Model Safety,
>   Mu et al., 2024
>
> On the Limited Generalization Capability of the Implicit Reward Model Induced by Direct Preference Optimization,
>   Lin et al., 2024
>
> RewardBench: Evaluating Reward Models for Language Modeling,
>   Lambert et al., 2024

---

> > ### Comment · Reviewer_ojiV · 2024-11-25
> > **Post-rebuttal**
> >
> > Thanks for the response. After reading the reviews, I found the shared novelty concerns and problems with clarity. Though the authors have addressed some of my concerns like the experiment setting, I still think there is room to improve this work. I tend to hold my original rating.

---

> ### Author Response · Authors · 2024-11-25
> **Thanks to Reviewer Feedback**
>
> Thank you for your constructive feedback. Can we make two more points that hope to clarify your novelty concern and our scientific contribution?
>
> 1. We have made significant updates to the paper, including the introduction of the *Strong-over-Weak Hypothesis*, which highlights the relationship between model choices and reward generalization. This finding could be particularly relevant to the community grappling with inconsistent or suboptimal outcomes when using DPO implicit rewards—a special case of density ratio rewards. Our analysis suggests that these challenges may be mitigated by the insights presented in our work. Result is illustrated in Figure-1, hoping you may find compelling and worthy of consideration.
>
> 2. I’d like to highlight again the novelty and impact of our proposed method: it is the first to employ prompts to customize density ratios as reward signals, leading to results that exceed conventional expectations. It points to the direction of customizing density ratios as untrained, off-the-shelf reward signal, and we are building a library to make its usage simple and accessible.

---

### Official Review · Reviewer_kBeT · 2024-11-04

**Soundness:** 3
**Presentation:** 3
**Contribution:** 3
**Rating:** 6
**Confidence:** 3

**Summary:**

The paper proposes a scalable, training-free method for preference data annotation in large language models by using guided density ratios between pre-trained models. This approach, which employs domain-specific prompts and few-shot examples, enhances alignment in areas like safety and reasoning without requiring costly human feedback. The method achieves competitive results on benchmarks such as RewardBench and MT-Bench, demonstrating its potential as an effective and resource-efficient alternative for preference tuning in LLMs.

**Strengths:**

1. The results on RewardBench and other benchmarks demonstrate that the proposed method can compete with or even outperform larger, more resource-intensive models, making it a cost-effective solution for preference alignment.

2. The paper includes comprehensive experiments across multiple domains and benchmarks, providing robust evidence for the effectiveness of the method.

3. The paper introduces a scalable, training-free preference annotation method using density ratios, a concept that could significantly reduce the cost and time typically required for human feedback data collection.

**Weaknesses:**

1. While prompt-based guidance improves performance, designing effective prompts and in-context examples for different domains could be labor-intensive and may not generalize well across all tasks.

2. I think evaluation across different model size should be included. For example, including 7B, 13B, and 70B.

**Questions:**

1. Could the density ratio approach be adapted to work with multi-modal data, like vision language models / LLaVA, or is it inherently limited to language models?

---

> ### Author Response · Authors · 2024-11-20
> **Response to Reviewer Questions**
>
> We thank the reviewer for their interest in our work and their constructive feedback. To address common questions and clarify the draft revisions, we have included a **Response-to-All-Reviewers (Parts 1, 2, 3)**. Below are responses to the reviewer’s specific questions.
>
> ---
>
> ### **W1:**
> To address the concern that “prompts and in-context examples…could be labor-intensive,” we discuss the exact cost of the safety prompt presented in the paper. The safety prompt required less than one day of work by one author. This time included drafting and iterating the instructions on a held-out set and preparing a set of in-context examples for each domain to ensure diversity. Note that we did not expend extra effort on optimizing these examples to achieve the results presented in the paper. Further discussion and ablation studies on prompt and ICL design are provided in **Response-to-All-Reviewers (Parts 2, 3)** and Appendix C.
>
> The reviewer raises an excellent point regarding the limited cross-domain generalization of prompting (“prompts and in-context examples…may not generalize well across all tasks”), and we agree that domain mismatch could lead to sub-optimal reward performance. To address this, we propose a **domain router** in the annotation pipeline (end of Section 3.2). This router ensures that each data point is routed to the most relevant domain prompt for reward annotation. Our experiment (Table 2) compares the alignment results of the annotation pipeline *with* and *without* the domain router, showing that the domain-router approach (e.g., AlpacaEval 40.6) outperforms using a single fixed domain prompt (e.g., math/coding 30.0, safety 36.0).
>
> ---
>
> ### **W2:**
> We appreciate the reviewer’s suggestion to experiment with even larger models using our method. We have initiated experiments with `meta-llama/Meta-Llama-3-70B-Instruct`, but the significant computational overhead associated with sampling/annotation and training at this scale presents a challenge. We will report the results as soon as they are available and include them in the revised draft.
>
> Notably, our current experiment already demonstrates that two relatively weak Mistral-7B models, without further training, can align the Llama-8B model to achieve GPT-4 level performance on AlpacaEval2.0 and ArenaHard. It will be fascinating to see if a similar super-alignment phenomenon can be achieved with the Llama-70B model.
>
> ---
>
> ### **Q1:**
> We believe that our density ratio-based method has applications beyond language models and holds promise for vision-language models (VLMs). Technically, as long as a model can input a sequence of tokens (e.g., text or image patches) and output their likelihood, our method can be applied.
>
> Moreover, based on the **Strong-over-Weak hypothesis** outlined in **Response-to-All-Reviewers (Part 1)**, we hypothesize that in the VLM setting, if we can identify two models where their difference is driven by human preference for generated images and the gap is sufficient, the difference in their density estimations can provide a quality/preference signal. We look forward to seeing follow-up work exploring this direction.

---

> ### Comment · Reviewer_kBeT · 2024-11-25
> **Feedbacks on the Rebuttal**
>
> While the authors claim that the prompt design took "less than one day,", it does not address the inherent difficulty of iteratively refining prompts and in-context examples when handling diverse, ambiguous, or overlapping domains. This issue may limit the practical utility of their approach. This concern on designing these prompts share similarity with the other two reviewers.

---

> ### Author Response · Authors · 2024-11-25
> **Thanks to your Constructive Feedback**
>
> Thank you for your constructive feedback.
>
> We fully understand and appreciate your feedback on the prompting process, which may limit the use-case of the method.
>
> But we want to highlight that the findings of the result do not depend on the prompting process or techniques. Density ratio reward improves from 77.2 to 80.1 on RewardBench by using a simple instruction “You are a helpful AI assistant.” Our more complex iteration of prompting only aims to show the it can specialize the density ratio towards different domains. Most noteworthy, by time of publishing, prompted density ratio scores the highest on the Safety domain of RewardBench, which is quite surprising as it out-performs specially trained safety classifiers. We believe those results sheds light on the strength of customization that clearly describing the "criterions of preference" is key to achieve improve performance for density ratio rewards. To make our method more useful, we are also building a library to use existing instructions to guide density ratio reward for it to be more accessible to the community.
>
>
> Besides this, can we make two more points that hope to highlight our updated paper and contributions?
>
> We have made significant updates to the paper, especially the introduction of the Strong-over-Weak Hypothesis, which highlights the relationship between model choices and reward generalization. This finding could be particularly relevant to the community grappling with inconsistent or suboptimal outcomes when using DPO implicit rewards—a special case of density ratio rewards. Our analysis suggests that these challenges may be mitigated by the insights presented in our work. Result is illustrated in Figure-1, hoping you may find compelling and worthy of consideration.
>
> I’d like to highlight again the novelty of our proposed method: it is the first to employ prompts to customize density ratios as reward signals, leading to results that exceed conventional expectations. It points to the direction of customizing density ratios as untrained, off-the-shelf reward signal.

---

### Official Review · Reviewer_QnfE · 2024-11-10

**Soundness:** 2
**Presentation:** 2
**Contribution:** 1
**Rating:** 3
**Confidence:** 4

**Summary:**

The paper interprets the use of a strong v/s weak LLM as a human preference proxy. However, their main contribution is designing prompt templates with relevant domain specific exemplars (IC examples) and domain specific system prompts - to show that the resulting density ratio is more helpful.

**Strengths:**

The paper does a good job in explaining their contribution, and the examples are helpful. The direction of LLM as a preference / human proxy is topical and being investigated by various groups - which makes it aligned to the venue.

**Weaknesses:**

I find the work to be an incremental advancement over previous discussions on using LLMs as a reward proxy. The key contribution of the work is the manually designed prompt templates, T(x), which are composed of system and in-context examples - both of which are domain dependent. The work's key argument is to favor a domain specific prompt design that influences density ratio - with limited discussions on why this template must work.

Few questions to the authors :
1. What is the cost of designing these templates? How can one extend this work to an arbitrary domain where the preferences may not be "safety".
2. In line 048, authors say that human preferences can be complicated. How are the preferences anymore complicated or undefined when they they pen down the preferences as part of their system prompt. Can the authors point out to examples where these prompt templates helped with other user preferences not explicitly mentioned in the prompt.
3. Can the authors compare the cost of prompt design with collecting more feedback?
4. What are the guarantees that this prompt template won't work adversarially to the model. For instance, its possible that the system prompt conflicts with some preferences specified in the query. What are some alternatives to the system prompt that the authors considered? The authors provide some discussion in A.1, however, the following questions remain unanswered :  how did the authors generate SYSTEM1, 2... versions; what are the domain specific principles and how were those generated?; what other principles did the authors experiment with? Which principles, while "plausible" and "aligned" with typical human preferences did not yield expected results.
5. Many a times IC examples have hampered model's performance. The authors discuss ICL example selection strategy in A.1, but the discussion seems to to justify the use of IC examples in their setup anecdotally.
6. The argument for why a baseline calibration is needed (lines 205 - 211) seem anecdotal. Can the authors provide stronger theoretical / empirical result on what are the properties of this weaker model for such a calibration, and what must the relationship between weaker and stronger model be in the scope of their work.

**Questions:**

Please see Weaknesses section.

I believe that addition of answers to the above questions can help with a stronger submission.

---

> ### Author Response · Authors · 2024-11-20
> **Response to reviewer questions**
>
> We thank the reviewer for their interest in our work and for their constructive feedback. Based on your suggestions, we update the drafe and add experiments. The primary update involves highlighting our first contribution: proposing the **Strong-over-Weak density ratio reward framework** and including experiments to support our thesis. Additionally, we conduct a study on prompt design, details in **Response-to-All-Reviewers (Part 2,3)** , to address concerns about the labor cost of designing prompts.
>
> ---
>
> ### **Q1:**
> Our experiments cover various domains, including Chat, Safety, and Coding/Math, as shown in Table 1. We include details about our prompt design process in Section C and add ablation studies on in-context learning (ICL) examples for various domains in Section C.1. Our work is not solely focused on the safety domain but is intended to present a general framework for customizing density ratio rewards.
>
> ---
>
> ### **Q1 & Q3:**
>
> The ability to craft rules and principles to supervise AI models is indeed critical for safe and controllable AI development (Mu et al., 2024) and offers significantly greater value than relying solely on manual annotation efforts.
>
> To address the reviewer’s question, the safety prompt presented in Figure-3 requires less than one day of work by one author. This time included drafting and testing the instructions on a held-out set and preparing the set of in-context examples to ensure diversity. It’s worth noting that we did not expend additional effort optimizing these examples to obtain the results shown in the paper. See **Response-to-All-Reviewers (Part 2)** for detail.
>
> In contrast, manual annotation entail much higher costs:
> - **Collecting data and training a reward model**: Top-performing reward models (e.g., RM-Mistral-7B with a score of 80.4 and ArmoRM-Llama-3-8B with a score of 90.4 in Table 1) require approximately 570k diverse preference samples, costing around $570 ($1 per sample), in addition to the cost of training the reward model.
> - **Directly collecting human feedback data**: Our experiments annotated 1,920,000 samples of preference data (Best-of-32 sampling) using our method. Annotating this manually would require a team of professional annotators working for months, costing approximately $1.92M ($1 per sample).
>
> ---
>
> ### **Q2:**
> We agree with the reviewer that human preferences can be complex. As outlined in **Contribution #1** in the introduction, our method's first design choice is selecting appropriate strong and weak model pairs capable of capturing the complexity of human preferences. Prompts serve an auxiliary role, refining and guiding the density ratio signal.
>
> For instance, a vanilla density ratio (without prompting) achieves 77.2 on RewardBench after careful selection of model pairs, as shown in Table 1.
>
> We also observe cross-domain generalization:
> - Shown in ablation Section C, the simple prompt "You are a helpful assistant" improves performance on the Reasoning domain, despite not explicitly targeting it.
> - Similarly, the **Safety prompt + ICL** in Table 1 enhances performance in both the Safety and Reasoning domains. The system instruction acts as a prior, enabling LLMs to generalize across tasks and domains.
>
> ---
>
> ### **Q4 & Q5:**
>
> The system prompt and ICL examples establish principles and guardrails that the model must follow. In cases of conflict between preference instructions and user instructions, preference instructions (defined in the system prompt) take precedence. For example, if a user requests unsafe information conflicting with safety principles, responses that disregard safety principles are penalized, while safe responses are rewarded.
>
> The rules are human-defined principles and are intended to be as such. As AI creators, we should have the ability to supervise principles AI systems shall follow, adjusting them for various deployment scenarios.
>
> We write **Response-to-All-Reviewers (Part 3)** and Section C.1 to address your question on ablating prompt design. For instance, we do observe diminishing returns after incorporating more than three rules in safety prompt, and the fifth rule (as shown in Figure 6) led to performance declines, so it was not adopted.
>
> ---
>
> ### **Q6:**
> We appreciate the constructive feedback. We agree that studying the relationship between strong and weak models is crucial to clarifying and strengthening **Contribution #1**. In response, we have expanded this topic as the **Strong-over-Weak hypothesis**, providing additional experiments as empirical evidence. Please see **Response-to-All-Reviewers (Part 1)** for more details.
>
> ---
>
> ### **References:**
> - **Rule-Based Rewards for Language Model Safety**
>   Tong Mu et al., 2024, [Link](https://api.semanticscholar.org/CorpusID:273812284)
>
> - **RewardBench: Evaluating Reward Models for Language Modeling**
>   Lambert et al., 2024, [Link](https://api.semanticscholar.org/CorpusID:268537409)

---

### Author Response · Authors · 2024-11-19
**Response to ALL reviewers on Major Updates (Part-1)**

We thank all reviewers for their time and valuable feedback. We decided to update our draft with experiments added to address some common concerns of reviewers.

## Improved presentation and additional empirical study for Contribution #1 (model choices)

While all reviewers have pointed out our Contribution #2 (prompting), our Contribution #1 on strong-over-weak model choices with off-the-shelf models in density ratio reward seems to be not well-recognized.

#### **1. Consolidation of Contribution #1 as the *Strong-over-Weak Hypothesis***
- In response to Reviewer QnfE’s suggestion and to clarify the significance of Contribution #1, we acted on the advice to update our presentation and explicitly stated and framed it as the *Strong-over-Weak Hypothesis*:
  > "The density ratio reward requires an alignment gap between a strong and weak model."
  This hypothesis challenges the conventional understanding that density ratio rewards are solely the result of optimizing the DPO objective. Instead, we assert that the alignment gap is a critical determinant of reward performance, providing a new perspective on how density ratio signals are generated and utilized.

#### **2. Empirical Validation Supporting the Strong-over-Weak Hypothesis**
- To substantiate this hypothesis, we conducted controlled experiments, as shown in updated **Figure 1**, using 221 distinct model pairs trained with diverse objectives, including SFT, DPO, PPO, KTO, RRHF, ORPO, SimPO, IPO, and SLiC-HF.
  - **Findings**:
    The results reveal a strong correlation between the reward performance and the *Model Alignment Gap* (ArenaHard diff.) between the strong and weak models. This empirical observation underpins the *Strong-over-Weak Hypothesis*.
  - We further demonstrated this phenomenon in **Figure 2** via a case study involving iterative DPO checkpoints. Key insights include:
    - **Red-line cells**: When the starting reference model is already well-aligned, the DPO objective struggles to learn a generalizable implicit reward.
    - **Green-line cells**: Conversely, selecting a weaker aligned model (e.g., the Base model) as the denominator in the density ratio consistently yields the highest reward scores. This practical guidance aligns with the *Strong-over-Weak Hypothesis*.

#### **3. Practical Implications of Contribution #1**
- Contribution #1 offers actionable insights for identifying successful model pairs when constructing density ratios, providing clear, experimentally validated criteria.
- Our thesis diverges from conventional beliefs that attribute density ratio rewards exclusively to DPO training. Instead, we demonstrate that these rewards are deeply influenced by the relative alignment between the models involved.
- This understanding also leads to novel practical benefits:
  - We introduce *prompt guidance* to exert control over the otherwise under-specified density ratio signal.
  - This stands in stark contrast to the implicit DPO thesis, where prompting an already optimal reward model would be counter-intuitive.

---

### Author Response · Authors · 2024-11-19
**Response to ALL reviewers on Major Updates (Part 2)**

## Ablation on Prompt Design and ICL Examples

We appreciate the reviewers’ comments regarding the labor cost involved in obtaining the current prompts and in-context learning (ICL) examples. This is a great question, and we updated our paper to highlight our finding that **density ratio** approach can benefit significantly from **domain-specific instructions**, even with limited effort in prompt engineering.

In response to reviewers KBeT, QnFE, and ojiV’s uniform suggestions, we conducted an ablation study on both the **rules and principles** (Appendix C) and the **ICL examples** used (Appendix C.1). Below, we summarize our findings.

---
### Key Takeaways

1. **Simple instructions** already bring noticeable improvements in reward performance.
2. Various **principles** we tested provide comparable performance and are straightforward to implement.
3. The **ICL examples** we used are not optimized yet still yield significant performance gains, leaving room for further improvements with more advanced techniques.

---

### Ablation on Rules and Principles (Section C)

Using the simple prompt, `"You are a helpful AI assistant."`, we observe a **2.9-point improvement** in the RewardBench score compared to the baseline without any prompt. Interestingly, the largest gains occur in the **Reasoning** domain, which includes tasks such as coding and math.

To better understand the impact of safety principles in prompt design, we conducted an iterative ablation study:

- **safe1**: Includes the first safety guideline.
- **safe2**: Adds a second guideline.
- **safe3**: Adds a third guideline.
- **safe4**: Incorporates all four guidelines.
- **safe5**: Adds a fifth guideline.

#### Observations
1. Adding the first few guidelines (safe1–safe3) consistently improves **Safety** scores.
2. Including the fourth guideline (**safe4**) results in diminishing returns and slight regressions in domains like **Reasoning**.
3. Adding a fifth guideline (**safe5**) leads to performance degradation, indicating that overloading prompts with rules can reduce effectiveness.

**Final Selection**: We chose **safe4** for its balance of comprehensive safety coverage and cross-domain performance. However, leaner prompts like **safe2** or **safe3** deliver comparable results for safety-focused tasks.

| **Prompt**           | **Chat** | **ChatHard** | **Safety** | **Reasoning** | **Overall** |
|-----------------------|----------|--------------|------------|---------------|-------------|
| -                     | 92.2     | 60.5         | 82.4       | 73.8          | 77.2        |
| simple                | 91.1     | 60.8         | 83.5       | **87.8**      | 80.1        |
| safe1                 | 93.8     | 56.8         | 83.9       | 81.2          | 79.0        |
| safe2                 | **94.7** | 57.7         | 89.3       | 82.6          | 81.1        |
| safe3                 | 93.0     | 60.1         | 90.2       | 82.4          | 81.7        |
| safe4                 | 91.1     | 59.2         | **91.6**   | 77.6          | 79.9        |
| safe5                 | 89.4     | 55.9         | 87.8       | 74.9          | 77.0        |

---

---

### Author Response · Authors · 2024-11-19
**Response to ALL reviewers on Major Updates (Part 3)**

### Ablation on the Effect of ICL Examples (Section C.1)

We designed ICL examples grouped by their primary domain (e.g., ChatHard, Safety, Math/Coding/Reasoning). While many examples overlap domains, each was classified by its primary focus for simplicity.

#### Observations
1. **Performance Increases**: All ICL examples improve density ratio reward performance on RewardBench.
2. **Minimal Differences**: Variations in performance between examples are minor and could result from noise or overfitting to the evaluation set of 2,850 examples.
3. **Simplicity**: Our ICL examples, written based on common design conventions, are unoptimized. This highlights the potential for further gains with advanced techniques.

**Summary of Results**:

| **ICL Example**        | **Chat** | **ChatHard** | **Safety** | **Reasoning** | **Overall** |
|-------------------------|----------|--------------|------------|---------------|-------------|
| -                       | 92.2     | 60.5         | 82.4       | 73.8          | 77.2        |
| **sys+ChatHard**        |          |              |            |               |             |
| ChatHard1               | 91.1     | 69.1         | 88.0       | 85.9          | 83.5        |
| ChatHard2               | 93.0     | 63.6         | 88.7       | 88.2          | 83.4        |
| ChatHard3               | 88.8     | 69.3         | 88.7       | 87.2          | 83.5        |
| ChatHard4               | 89.9     | 66.0         | 91.9       | 86.6          | 83.6        |
| ChatHard5               | 90.5     | 63.8         | 91.7       | 89.7          | 83.9        |
| ChatHard6               | 94.7     | 59.9         | 89.2       | 89.3          | 83.4        |
| **sys+Safety**          |          |              |            |               |             |
| Safe1                   | 88.3     | 61.8         | 91.0       | 87.9          | 82.3        |
| Safe2                   | 90.8     | 64.3         | 89.7       | 86.2          | 82.8        |
| Safe3                   | 91.3     | 60.1         | 91.1       | 87.8          | 82.6        |
| **sys+Reasoning**       |          |              |            |               |             |
| Reasoning1              | 91.9     | 59.9         | 90.1       | 88.7          | 82.7        |
| Reasoning2              | 91.9     | 61.2         | 88.2       | 87.0          | 82.1        |
| Reasoning3              | 90.2     | 64.3         | 90.0       | 85.8          | 82.6        |
| Reasoning4              | 90.5     | 61.8         | 89.5       | 88.7          | 82.6        |
| Reasoning5              | 93.6     | 61.6         | 88.7       | 87.1          | 82.8        |
| Reasoning6              | 91.6     | 58.8         | 88.8       | 87.5          | 81.7        |
| Reasoning7              | 88.3     | 60.1         | 89.9       | 87.0          | 81.8        |
| Reasoning8              | 91.6     | 61.0         | 89.9       | 89.7          | 83.1        |

---

### Conclusion
We hope above ablations can address reviewer concern on the labor cost of prompt design.



### Summary of Revisions to Draft

1. **Abstract and Introduction**:
   - Updated to frame and clearly present the **Strong-over-Weak Hypothesis**, providing a compelling rationale for our approach.

2. **Method Section**:
   - Revised to emphasize the **Strong-over-Weak reward function design**, detailing its role as the core innovation of our methodology.

3. **Experiment Section**:
   - Enhanced to validate the **Strong-over-Weak Hypothesis** with supporting empirical evidence.
   - Clarified the experiment design to ensure transparency and reproducibility of results.

4. **Appendix**:
   - Expanded to include an **Ablation on Prompt Design**, offering deeper insights into the effects of various prompts.
   - Added more detailed examples of the prompts used in experiments to aid understanding and replication.

---

### Meta-Review · Area_Chair_794j · 2024-12-25

**Metareview:**

The submission addresses the problem of preference tunning in large language models, by introducing a novel data annotation technique based on guided density ratios between pre-trained models. The proposed method achieves competitive performance on RewardBench and MT-Bench. The submission received mixed ratings after rebuttal, including one borderline accept (6), one borderline reject (5), and one reject (3). Reviewer QnfE who gave reject did not engage in the post-rebuttal discussion. Below I summarize the main strengths and limitations of the submission, according to the reviews (after rebuttal discussion) and my own reading of the submission:

*Strengths:*
- The proposed method is scalable and cost efficient, and is of practical importance.
- The submission offers extensive empirical evaluations, demonstrating the effectiveness of the proposed approach.

*Weaknesses:*
- The proposed method employs domain-specific prompts, which could be labor-intensive and may not generalize well (QnfE, kBeT, ojiV).
- Lack of clarity on the submission's claimed contributions (QnfE), and that the proposed prompt template design has been explored (ojiV).

After the rebuttal, the AC shares QnfE and kBeT's remaining concern on the practicality of the prompt design used by the proposed approach, and ojiV's concern on the clarity and novelty of the proposed approach. There was no reviewer championing for the acceptance of the submission. The AC thus believes that the submission is not ready to be accepted.

**Additional Comments On Reviewer Discussion:**

Please find above.

---

### Decision · Program_Chairs · 2025-01-22

Reject